# Gel Synthesis of Hexaferrites Pb_1−*x*_La*_x_*Fe_12−*x*_Zn*_x_*O_19_ and Properties of Multiferroic Composite Ceramics PZT–Pb_1−*x*_La*_x_*Fe_12−*x*_Zn*_x_*O_19_

**DOI:** 10.3390/nano10091630

**Published:** 2020-08-19

**Authors:** Inna V. Lisnevskaya, Inga A. Aleksandrova

**Affiliations:** Faculty of Chemistry, Southern Federal University, 344090 Rostov on Don, Russia; ing_a@ro.ru

**Keywords:** multiferroics, magnetoelectric composites, hexaferrites, gel synthesis

## Abstract

We investigated the opportunities for obtaining hexaferrites Pb_1−*x*_La_*x*_Fe_12−*x*_Zn*_x_*O_19_ (*x* = 0–1) from citrate–glycerin gel and showed that synthesis occurs via the formation of the Fe_3_O_4_ phase; products with a small amount of hematite impurity Fe_2_O_3_ can be obtained after firing at 800 to 900 °C with 0 ≤ *x* ≤ 0.5. If *x* > 0.5, perovskite-like LaFeO_3_ is formed in samples, so that if *x* = 0.9–1, the synthesis products virtually do not contain phases with hexaferrite structures and represent a mixture of LaFeO_3_, Fe_2_O_3_, and Fe_3_O_4_. Within the range of 0 ≤ *x* ≤ 0.5, the electrical and magnetic characteristics of hexaferrites Pb_1−*x*_La*_x_*Fe_12−*x*_Zn_*x*_O_19_ are slightly dependent on *x* and have the following average values: A relative permittivity ε/ε_0_ ~ 45, a dielectric loss tangent tan δ ~ 0.6, an electrical resistivity R ~ 10^9^ Ohm cm, coercivity H_c_ ~ 3 kOe, saturation magnetization M_s_ ~ 50 emu/g, and remanent magnetization M_r_ ~ 25 emu/g. The magnetoelectric (ME) ceramics 50 wt.% PZTNB-1 + 50 wt.% Pb_1−*x*_La*_x_*Fe_12−*x*_Zn_*x*_O_19_ (PZTNB-1 is an industrial piezoelectric material based on lead titanate zirconate (PZT) do not contain impurity phases and have the following characteristics: Piezoelectric coefficients d_33_ = 10–60 and −d_31_ = 2–30 pC/N, piezoelectric voltage coefficients g_33_ = 2–13 and −g_31_ = 1–5 mV m/N, an electromechanical coupling coefficient K_p_ = 0.03–0.13, magnetic parameters H_c_ = 3–1 kOe, M_s_ = 50–30, and M_r_ = 25–12 emu/g. The maximum ME coupling coefficient ΔE/ΔH ~ 1.75 mV/(cm Oe) was achieved with *x* = 0.5.

## 1. Introduction

Ferrites with a structure of magnetoplumbite MFe_12_O_19_ (where M = Ba, Sr, Pb, and other large doubly charged cations) are long- and well-known [1,2,3]. They are magnetically hard materials with high coercivity, high saturation magnetization, and large magnetic anisotropy. Hexaferrite ceramics are used as permanent magnets, in microwave sources, and in the production of high-density magnetic storage media. Their advantages are high resistivity and low cost [4].

In the hexaferrite family, special attention is drawn to lead ferrite as a hard magnetic material with relatively low synthesis and sintering temperatures. The structure of PbFe_12_O_19_ is shown in Figure 1a. Like the structure of other hexaferrites with a magnetoplumbite structure, it can be described as a combination of spinel and hexagonal blocks. In this case, oxygen anions, together with large lead ions, are densely packed with an alternation sequence of 10 ABABACBCBC layers along the *c* axis, where the third and eighth layers contain Pb^2+^ cations. One unit cell contains two formula units of PbFe_12_O_19_. The first coordination sphere of Pb^2+^ cations has the form of an anticuboctahedron (Figure 1b), whereas iron cations are located at five nonequivalent crystallographic positions (Figure 1c–g): Fe1 (2a), Fe4 (4f1), and Fe5 (12k) and have an octahedral, Fe2 (2b) and Fe3 (4f2), respectively, trigonal-bipyramidal, and tetrahedral first oxygen coordination sphere. According to the crystal field theory, iron cations have no coordination preferences. As for Fe^3+^ cations, the crystal field stabilization energy (CFSE) is equal to zero in the field of any symmetry (Figure 2). Below the Curie temperature, iron cations form five magnetic sublattices. Magnetic moments of iron ions in crystallographic positions 12k, 2a, and 2b make the main contribution to the total magnetization, whereas the contribution of iron cations located at positions 4f1 and 4f2 is the inverse.

Recently, it has emerged that ferrites with magnetoplumbite structures have multiferroic properties, i.e., they demonstrate an interconnection of electrical and magnetic parameters at temperatures above room temperature, which makes them promising for a new range of various practical applications. Kostishin et al. [5] showed that the polycrystalline hexaferrites BaFe_12_O_19_ and SrFe_12_O_19_ have an improved set of properties: Their remanent magnetization, as well as magnetoelectric and magnetodielectric constants, is several times higher than those of the most well-known high-temperature single-phase multiferroic BiFeO_3_ in the thin film version, which was first obtained by Wang et al. [6], with similar values of maximum electric polarization. The properties of PbFe_12_O_19_ obtained in the paper by Kostishin et al. [5] are noticeably worse than those of barium and strontium hexaferrites; however, it is possible to improve them, as shown in the paper by Tan et al. [7], through firing the samples in oxygen. The results of the PbFe_12_O_19_ study obtained by Tan et al. in [8] also confirm its dual ferrimagnetic/ferroelectric nature and advantageous characteristics compared to BiFeO_3_ ceramics.

In papers by Tan et al. [7,8], as well as in a number of other publications [9,10,11,12,13], lead hexaferrite was synthesized using “wet” chemistry methods. The use of low-temperature methods for its synthesis is justified by the need to preserve the final product stoichiometry, which may be disturbed during heat treatment due to the volatility of lead oxide. According to one of the earliest of the above mentioned studies by Gajbhiye et al. [9], single-phase lead hexaferrite can be obtained from citrate gel after firing at 500 to 800 °C; however, unfortunately, the authors provided no diffraction pattern of the obtained products. Ghahfarokhi et al. [10] have shown that obtaining the desired product requires firing at a temperature of 850 °C; however, it contains an impurity of α-Fe_2_O_3_ (~8%). At lower (700 to 800 °C), as well as at higher (900 to 1000 °C) temperatures, the amount of hematite in the samples is significantly higher and reaches ~20–50%. The lead hexaferrite samples obtained by Mahdiani et al. [11] at 900 °C, in addition to Fe_2_O_3_, contain an impurity of Fe_3_O_4_ magnetite, and the use of a long ultrasonic treatment at maximum power in the early synthesis stages helps to reduce the content of impurity phases. Finally, according to Mahdiani et al. [11], Yang et al. [12], and Rostami et al. [13], a deviation of the Pb:Fe ratio from the stoichiometric one towards the increase of lead has a beneficial effect on the final product purity.

There are a number of studies that consider hexaferrites as potential magnetostrictive components of two-phase multiferroic systems (magnetoelectric (ME) composite materials). An obvious advantage of such composites, unlike, for example, similar systems based on soft magnetic ferrites (spinels or garnets), is the absence of necessity to create a magnetizing field to implement the ME effect, as hexaferrite-based composite ceramics, due to their magnetic hardness, can be pre-magnetized and, therefore, can maintain their own remanent magnetization if they are operated in alternating magnetic fields of low strength, which is not sufficient for demagnetization.

In the literature, there is a lot of data about biphasic multiferroic systems based on hexaferrites of strontium [14,15] and barium [16,17,18,19,20,21,22], as well as solid solutions based on BaFe_12_O_19_ (BaFe_12−*x*_Mn*_x_*O_19_ [23], BaFe_11.9_Al_0.1_O_19_ [24]), where barium titanate [14,15,16,20,21,22,23,24], strontium barium titanate [19], bismuth sodium titanate [18,21,22], and potassium–sodium niobate [17] are used as piezoelectric components. The authors of most of the above mentioned papers note the absence of extraneous phases in the composites after firing, although the composites described by Want et al. [20] and by Trukhanov et al. [24] contain impurity phases with a spinel structure. With an increase in the ferrite content, there is a regular decrease in the dielectric permittivity, remanent electric polarization, and coercive force, and an increase in the remanent magnetization as described by Rather et al. [14], Pular et el. [15], Kumar et al. [17], and by Pattanayak et al. [18]. We can also see the dependence of these parameters on the phase particle sizes in the content of the composites [24]. Depending on their compositions, the electrical properties of the composites often change nonlinearly; apparently, there are percolation effects [15,18]. Gao et al. [23] have shown that barium hexaferrite doping with manganese cations provides an increase in the resistivity of composite ceramics and a decrease in the dielectric loss tangent. The hexaferrite-based composites exhibit multiferroic properties. Thus, 0.4–0.2BaFe_12_O_19_/0.6–0.8BiTiO_3_ composite ceramics (Srinivas et al. [16]) demonstrate the ME effect of 0.14 mV/(cm·Oe); in 0.3BaFe_12_O_19_/0.7Na_0.5_K_0.5_NbO_3_ composites (Kumar et al. [17]) it is noticeably higher and reaches 4.08 mV/(cm·Oe). BaFe_12_O_19_/Ba_0.5_Sr_0.5_TiO_3_ heterostructures (Das et al. [19]), grown on a sapphire substrate by pulsed laser deposition, demonstrate shift of the line of ferromagnetic resonance of ~11 Oe when applying 9 V to the ferroelectric layer, which corresponds to a tunability of 3.5 MHz/V at 60 GHz. In a two-phase system of 0.9BaY_0.025_Ti_0.9625_O_3_/0.1SrFe_12_O_19_ (Dather et al. [14]), a slight increase in magnetoresistance (0.03%) can be observed at a frequency of 1 kHz, which the authors interpret as a manifestation of the magnetodielectric effect. Unfortunately, most studies provide no data on the ME properties of the composites.

The paper by Jaffari et al. [25] is one of the few publications that describes ME composite ceramics based on lead hexaferrite. Structural, optical, ferroelectric, spectroscopic, and magnetic studies of the properties of the composites PbTi_1−*x*_Fe_*x*_O_3_/PbFe_12_O_19_ depending on the iron concentration were performed. Evidence of the existence of ME coupling in these composites was the recording of an unusual temperature dependence of the magnetic coercivity.

The aim of this study was to investigate the possibilities of obtaining solid solutions based on lead hexaferrite Pb_1−*x*_La*_x_*Fe_12−*x*_Zn*_x_*O_19_ using the gel method, as well as obtaining and studying the properties of magnetoelectric (ME) composites 50 wt.% PZTNB-1 + 50 wt.% Pb_1−*x*_La*_x_*Fe_12−*x*_Zn_*x*_O_19_, where piezoelectric material based on lead titanate zirconate (PZT) of the PZTNB-1 industrial brand is used as a piezoelectric component. This material was previously successfully used in the composites based on modified nickel ferrite [26,27,28,29,30] and yttrium iron garnet [31] and has the following parameters: An electromechanical coupling coefficient of K_p_ = 0.6, a permittivity of ε_33_^T^/ε_0_ = 1400, a piezoelectric coefficient d_33_ = 330 pC/N, and a piezoelectric voltage coefficient g_33_ = 27 mV m/N. Materials of the PZT system have actively been used as piezoelectric components of ME composites in a number of studies, for example, in Pb(Zr,Ti)O_3_/CoFe_2_O_4_ systems, studied by Kotnana et al. [32] and by Li et al. [33], and in multiferroic materials based on polymer, NdFeB, and PZT particles, studied by Makarova et al. in [34]. As stated above, the use of low-temperature methods in the synthesis of Pb_1−*x*_La*_x_*Fe_12−*x*_Zn*_x_*O_19_ is relevant due to the volatility of lead oxide. Doping of lead ferrite with Zn^2+^ cations has been studied, as a number of doubly charged cations of d-elements may have the most pronounced coordination preferences for tetrahedral positions of the magnetoplumbite structure, which can have a strong effect on the magnetic parameters of hexaferrites; in this case, La^3+^ cations were introduced for charge compensation. The effect of heterovalent substitutions with Zn^2+^/La^3+^ cations on the properties of barium hexaferrite was investigated in papers by Gruskova et al. [35] and Kumar et al. [36], and it has been shown that, in the Ba_1−*x*_La*_x_*Fe_12−*x*_Zn*_x_*O_19_ system, within the range of *x* = 0–0.4, there is a continuous series of solid solutions. According to [26], there is a slight decrease in the Curie temperature and the coercive force and an increase in the saturation magnetization M_s_ and the remanent magnetization M_r_. According to [27], the samples demonstrate multiferroic properties. There is spontaneous electric polarization, reaching the highest values at *x* = 0.2, for which the magnetization M_s_ and M_r_ reaches the maximum values (40 and 24 emu/g, respectively). The samples demonstrate the ME effect with the ME coupling coefficient ~1.69 × 10^−6^ mV/(cm Oe). All this suggests that investigating a similar system Pb_1−*x*_La*_x_*Fe_12−*x*_Zn*_x_*O_19_ may be of interest; however, there is no data on it in the literature.

## 2. Materials and Methods 

To obtain solid solutions of Pb_1−*x*_La*_x_*Fe_12−*x*_Zn*_x_*O_19_, where *x* = 0–1, the required amount of ferric citrate (Reagent Inc., Ufa, Russia) was weighed, the iron content was previously measured using the gravimetric method, and then it was dissolved in water with citric acid and glycerin. Solutions of nitrates of lead, lanthanum, and zinc were separately prepared by dissolving the necessary amounts of oxides PbO (Nevchimik Inc., Saint Petersburg, Russia), La_2_O_3_ (Henan Kingway Chemicals Co., Ltd., Shanghai, China), and ZnO (Nevchimik Inc., Saint Petersburg, Russia), in nitric acid (Yugkhimprodukt, Rostov-on-Don, Russia), where the calculated amounts of citric acid (Yugreactiv Ltd., Rostov-on-Don, Russia) and glycerin (Yugreactiv Ltd., Rostov-on-Don, Russia) were then introduced. Lead oxide was taken in excess of 5 wt.% to compensate for losses during the heat treatment. Molar ratios of cation:citric acid:polyhydric alcohol in the final solution were 1:3:1, in accordance with the overall reaction equation:(1−*x*) Pb^2+^ + *x* La^3+^ + (12−*x*) Fe^3+^ + *x* Zn^2+^ + 39 H_7_C_6_O_7_^−^ + 13 C_3_H_6_O_3_ + 214.25 O_2_ →→ Pb_1−*x*_La*_x_*Fe_12−*x*_Zn*_x_*O_19_ + 273 CO_2_ + 175.5 H_2_O.(1)

The resulting solutions were poured together and slowly evaporated at a temperature of ~90 °C; during evaporation, transparent polymer gels were obtained. Then, they were dried by slowly raising the temperature to ~120 °C. To study the electrophysical and magnetic properties of Pb_1−*x*_La*_x_*Fe_12−*x*_Zn*_x_*O_19_ solid solutions, xerogels were calcined at 800 °C for 3 h; after that, ceramic samples were sintered from the obtained ferrite powders at a temperature of 1000 °C for 2 h, from which discs with a height of 1 mm were made for further measurements.

To study the phase formation processes, Pb_1−*x*_La*_x_*Fe_12−*x*_Zn*_x_*O_19_ xerogels were calcined at 200 to 1000 °C with a step of 100 °C for 1.5–2 hours and the obtained powders were studied using the X-ray powder diffraction analysis (XRD) method on an ARL-X’TRA diffractometer (Thermo Fisher Scientific, Waltham, MA, USA) in CuK_α_ radiation. Pb_1−*x*_La*_x_*Fe_12−*x*_Zn*_x_*O_19_ xerogels were also studied using the methods of differential scanning calorimetry (DSC) and thermogravimetric analysis (TGA) on a NETZSCH STA 449 C derivatograph (NETZSCH Holding, Selb, Germany). The samples were heated in air in corundum crucibles at a rate of 5°/min in the temperature range from 100 to 900 °C.

The obtained hexaferrites were tested as magnetostrictive phases for the preparation of ME composite ceramics 50 wt.% PZTNB-1 + 50 wt.% Pb_1−*x*_La*_x_*Fe_12−*x*_Zn*_x_*O_19_. To obtain composites from ferrite powders and piezoelectric material, pressed blanks, which were sintered at a temperature of 1000 °C for 2 hours, were prepared. To control the phase composition of composite ceramics, the XRD method was used.

From the sintered ceramic samples 50 wt.% PZTNB-1 + 50 wt.% Pb_1−*x*_La*_x_*Fe_12−*x*_Zn*_x_*O_19_, tablets 1 mm in height were made, on which electrodes were applied by burning silver-containing paste at 600 °C. The elements were polarized in a pulsed mode in a carbon tetrachloride medium with a field of 3–4 kV/mm at room temperature for 3 min.

The morphology of ME ceramics in chipped areas was studied by electron microscopy (a focused-beam electron microscope JEOL JSM 6390LA (International Equipment Trading Ltd., Mundelein, IL, USA).

Dielectric and piezoelectric properties of ME composite ceramics were studied using a computerized measuring system “Tsenzurka M”. The piezoelectric coefficient d_33_ was measured using the quasi-static method with a d_33_ meter. The resistivity of the composites was measured using a DC tera-ohm meter E6 13A (Priborteh, Moscow, Russia).

The magnetic parameters of the samples of Pb_1−*x*_La*_x_*Fe_12−*x*_Zn*_x_*O_19_ ferrites and the composite ceramics on their basis were measured using a LakeShore VSM 7404 (LakeShore Cryotronics Inc., Westerville, OH, USA) vibration magnetometer at room temperature.

The morphologic characteristics of the material were determined using the method of transmission electron microscopy (TEM) using a microscope TEM Tecnai G2 Spirit Bio TWIN (FEI Company, Hillsboro, OR, USA).

The linear ME effect of composite ceramics was studied on the pre-magnetized samples at a frequency of 1 kHz under conditions of convergence of vector directions of electric polarization and the ME ceramics magnetization and strength of the alternating magnetic field (19.2 Oe).

## 3. Results and Discussion

The gel method for obtaining Pb_1−*x*_La*_x_*Fe_12−*x*_Zn*_x_*O_19_ hexaferrites is based on the well-known idea of Pechini [37], consisting of performing esterification reactions of metal chelate complexes with polyhydric alcohols with the formation of polymer gels. Schematically, the citrate glycerin gel formation process is shown in Figure 3. It is believed that the formation of chelate metal complexes and the subsequent polycondensation reactions level the difference in the individual cation behavior in the solution, which contributes to a more complete mixing and prevents separate crystallization processes at subsequent synthesis stages.

Figure 4 shows the TGA and DSC curves of the xerogel obtained during the synthesis of PbFe_12_O_19_ (for solid solutions of Pb_1−*x*_La*_x_*Fe_12−*x*_Zn*_x_*O_19_, the graphs are basically the same). It can be seen that the oxidation processes of organic residues of xerogels under the influence of atmospheric oxygen were completed at ~450 °C, and within the temperature range from ~250 to 450 °C, the mass of the samples decreased rapidly, which was accompanied by a pronounced exothermic effect. Above 450 °C, there were no changes in the TGA and DSC curves.

According to the X-ray powder diffraction analysis, within the temperature range of 200 to 500 °C, the gel synthesis of PbFe_12_O_19_ hexaferrite and solid solutions of Pb_1−*x*_La*_x_*Fe_12−*x*_Zn*_x_*O_19_ irrespective of *x* passed through the stage of Fe_3_O_4_ magnetite formation with a spinel structure, as shown in Figure 5a in the example of the composition with *x* = 0. Fe_3_O_4_ contains iron in a mixed oxidation state of +2 and +3. The xerogel-reducing medium probably help stabilize this phase where, according to the differential thermal analysis, organic residues were present up to a temperature of ~450 °C. The formation of phases with a hexaferrite structure started at a temperature of 600 °C. At this stage of the synthesis, the samples contained the impurity phases—Fe_2_O_3_ with a hematite structure and PbO in a massicot modification. Their highest peaks can be observed on the diffraction patterns of the synthesis products. PbO reflection intensities were extremely low (in Figure 5, and they are indicated by arrows), which was naturally due to a small fraction of lead in the samples, in accordance with their stoichiometry. The reason for the presence of the impurity phases in the synthesis of PbFe_12_O_19_ and solid solutions of Pb_1−*x*_La*_x_*Fe_12−*x*_Zn_x_O_19_ can be seen in the fact that, during their synthesis using the gel method, the spinel phase was formed at the early stages, whereas the synthesis of the desired phases with a hexaferrite structure was performed in the solid-phase mode, when lead oxide was present in the system as a separate phase, and diffusion processes were hindered. With an increase in the firing temperature to 800–900 °C, the reflection intensity of the impurity phases decreased; however, even after isothermal exposure at 900 to 1000 °C, it was not possible to completely eliminate Fe_2_O_3_ impurities in the sample with *x* = 0–0.5. For *x* < 0.5, Fe_2_O_3_ was the only impurity phase observed. At *x* = 0.5, there were barely any impurities of the perovskite-like phase of LaFeO_3_ in the Pb_1−*x*_La*_x_*Fe_12−*x*_Zn*_x_*O_19_ synthesis products, while at *x* = 0.7, it became clearly distinguishable. Finally, the samples with *x* = 0.9–1 practically did not contain phases with the hexaferrite structure and represented a mixture LaFeO_3_, Fe_2_O_3_, and Fe_3_O_4_.

In the existence range of the products of Pb_1−*x*_La*_x_*Fe_12−*x*_Zn*_x_*O_19_ (0 ≤ *x* ≤ 0.5) with a low content of impurity Fe_2_O_3_, their electrical and magnetic characteristics were weakly dependent on *x* and had the following averaged values: A relative permittivity ε/ε_0_ ~ 45, a dielectric loss tangent tanδ ~ 0.6, a resistivity R ~ 10^9^ Ohm cm, coercivity H_c_ ~ 3 kOe, saturation magnetization M_s_ ~ 50 emu/g, and remanent magnetization M_r_ ~ 25 emu/g (see accurate values and details in Appendix A). Figure 6 shows TEM images of Pb_1−*x*_La*_x_*Fe_12−*x*_Zn*_x_*O_19_ (*x* = 0.1) nanoparticles synthesized by gel processing (for TEM images for samples with other values of *x,* see in Appendix A). The average particle size of Pb_1−_*_x_*La*_x_*Fe_12−*x*_Zn*_x_*O_19_ powders after synthesis was ~100 nm.

The results for the composite ceramics 50 wt.% PZTNB-1 + 50 wt.% Pb_1−*x*_La*_x_*Fe_12−*x*_Zn*_x_*O_19_ based on hexaferrites with *x* = 0–0.5 should be discussed. Figure 7 shows the diffraction patterns of the hexaferrite composite with *x* = 0.1 (diffraction patterns of other composites are essentially the same (see details in Appendix A), as well as the X-ray profiles of the phases constituting the composite). It can be seen that composite ceramics consisted of two phases: Perovskite (PZTNB-1) and magnetoplumbite (Pb_1−*x*_La*_x_*Fe_12−*x*_Zn*_x_*O_19_). No extraneous phases were detected; moreover, those reflections that belonged to a small hematite impurity in ferrites after the synthesis were not found in composite ceramics, which can be a sign of the piezoelectric doping-phase process during high-temperature firing of the samples. At the same time, phase reflections in the content of the composites were not shifted relative to the pure component peaks; therefore, it can be assumed that interfacial doping occurred only along the grain boundaries.

Figure 8a,c shows SEM images of the microstructure of the hexaferrite-based composites with various values of *x*. As it can be seen, regardless of the hexaferrite composition, composite ceramics have the same structural features: They have a dense structure and consist of large piezoelectric grains with sizes of 2–10 μm and larger in the space between which there are small grains of ferrite of less than 2 μm in size. The characteristic hexagonal shape of the largest crystallites of the hexaferrite phase (see insertions on Figure 8), as well as the absence of their preferential orientation in the ceramic samples, can be noted.

Figure 9a–c shows the concentration dependence of the dielectric permittivity ε_33_^T^/ε_0_, the resistivity logarithm lgR, and the dielectric loss tangent tanδ of composite ceramics 50 wt.% PZTNB-1 + 50 wt.% Pb_1−*x*_La*_x_*Fe_12−*x*_Zn*_x_*O_19_. It can be seen that introducing dopants generally improved the electrical properties of the composites, decreasing the direct current conductivity and dielectric losses in alternating fields. At the same time, the electrical resistance of the composites (10^6^–10^8^ Ohm cm) was significantly lower than that of the pure phases (characteristic values ~10^9^ and ~10^11^ Ohm cm for ferrites and PZTNB-1, respectively), from which it can be concluded that the electrical conductivity of the samples was conditioned by the state of the interfacial boundaries, which had a low resistance due to interfacial doping processes.

Apparently, the same interfacial doping processes are the reason for a sharp dependence of the magnetic parameters of the ME composites 50 wt.% PZTNB-1 + 50 wt.% Pb_1−*x*_La*_x_*Fe_12−*x*_Zn*_x_*O_19_ on *x*, while the properties of ferrites themselves, on the basis of which they are made, are rather close to each other. Figure 10 shows magnetic hysteresis loops for composite ceramics 50 wt.% PZTNB-1 + 50 wt.% Pb_1−*x*_La*_x_*Fe_12−*x*_Zn*_x_*O_19_, as well as (for comparison) the form of the hysteresis loop for one of the ferrites (as stated above, magnetic characteristics of Pb_1−*x*_La*_x_*Fe_12−*x*_Zn*_x_*O_19_ ferrites within the range of *x* = 0–0.5 and, hence, the forms of hysteresis loops differ little from each other). It can be seen that, with an increase in the concentration of doping additives in the ferrite composition within the range of *x* = 0–0.5, there was an approximately two-fold decrease in the coercivity H_c_, the saturation magnetization M_s_, and the remanent magnetization M_r_ of the composites based on them; in other words, the magnetic softness of composite ceramics 50 wt.% PZTNB-1 + 50 wt.% Pb_1−_*_x_*La*_x_*Fe_12−*x*_Zn*_x_*O_19_ increased and, at the same time, ceramics still remained magnetically hard, because they had a high coercivity H_C_ > 1 kOe.

The concentration dependences of the piezoelectric parameters of the composites 50 wt.% PZTNB-1 + 50 wt.% Pb_1−*x*_La*_x_*Fe_12−*x*_Zn*_x_*O_19_—piezoelectric coefficients d_33_ and −d_31_, piezoelectric voltage coefficients g_33_ and −g_31_, and the longitudinal coefficient of electromechanical coupling K_p_—are shown in Figure 11a–c. With the same mass content of the composites, with increasing concentration of doping additives in the ferrite phase within the range of *x* = 0–0.3, there was a significant increase in piezoelectric coefficients d_33_ and −d_31_ and the coupling coefficient K_p_, and only when it came to the samples with *x* = 0.5 was there a slight decline in them. Piezoelectric voltage coefficients g_33_ and −g_31_, as the parameters depending simultaneously on d_33_ (−d_31_) and ε_33_^T^, increased over the entire range of *x*.

A significant improvement in the piezoelectric properties of the composites 50 wt.% PZTNB-1 + 50 wt.% Pb_1−*x*_La*_x_*Fe_12−*x*_Zn*_x_*O_19_, which can be observed with an increase in the content of doping components in the hexaferrite composition as well as a change in their magnetic parameters towards the increase of the magnetic softness, was an unexpected positive result, which can be associated with the interfacial doping processes. We can presumably give the following explanation for the discovered phenomena. As noted in the Introduction, below the Curie temperature, iron cations form five magnetic sublattices. Magnetic moments of iron ions in crystallographic positions 12k, 2a, and 2b made the main contribution to the total magnetization, whereas the contribution of iron cations located at positions 4f1 and 4f2 was the inverse. Doping of Pb_1−*x*_La*_x_*Fe_12−*x*_Zn*_x_*O_19_ with nonmagnetic cations that are part of PZT (Zr^4+^, Ti^4+^) and tend to occupy octahedral positions can strongly affect the magnetic properties of hexaferrites. In this case, the more doubly charged Zn^2+^ cations that were present in the hexaferrite composition, the more active the doping process was, because the charge of the mixed (Zn_1/2_(Ti,Zr)_1/2_)^3+^ cation was equal to the charge of the Fe^3+^ cation. As a consequence, this should facilitate the process of doping of the PZT phase with La^3+^ cations, which interacted with the Fe_2_O_3_ impurity (which, as noted, was absent in the composites, unlike the ferrites themselves) with the formation of perovskite-like LaFeO_3_, isostructural to PZT, which led to an improvement in the piezoparameters of the PZT phase as a part of the composites.

Figure 11d shows a graph of changes in the linear ME coefficient of the composite materials 50 wt.% PZTNB-1 + 50 wt.% Pb_1−*x*_La*_x_*Fe_12−*x*_Zn*_x_*O_19_, as measured on the pre-magnetized samples in the absence of an external constant magnetic field. The maximum value of the ME effect ΔE/ΔH ~ 1.75 mV/(cm Oe) was achieved on the most magnetically soft samples with *x* = 0.5, which had the maximum piezoelectric sensitivity among the composites considered in this paper, which seemed to be quite natural.

## 4. Conclusions

The opportunities for obtaining hexaferrites Pb_1−*x*_La*_x_*Fe_12−*x*_Zn*_x_*O_19_ (*x* = 0–1) from the citrate–glycerin gel were investigated and it was shown that synthesis occurred via the formation of the Fe_3_O_4_ phase; the products with a small amount of hematite impurity Fe_2_O_3_ could be obtained after firing at 800 to 900 °C with 0 ≤ *x* ≤ 0.5. If *x* > 0.5, perovskite-like LaFeO_3_ was formed in the samples, so that if *x* = 0.9–1, the synthesis products virtually did not contain the phases with hexaferrite structures and represented a mixture of LaFeO_3_, Fe_2_O_3_, and Fe_3_O_4_. Within the range of 0 ≤ *x* ≤ 0.5, the electrical and magnetic characteristics of hexaferrites Pb_1−*x*_La*_x_*Fe_12−*x*_Zn*_x_*O_19_ were slightly dependent on *x* and had the following averaged values: A relative permittivity ε/ε_0_ ~ 45, a dielectric loss tangent tanδ ~ 0.6, an electrical resistivity R ~ 10^9^ Ohm cm, coercivity H_c_ ~ 3 kOe, saturation magnetization M_s_ ~ 50 emu/g, and remanent magnetization M_r_ ~ 25 emu/g. The magnetoelectric (ME) ceramics 50 wt.% PZTNB-1 + 50 wt.% Pb_1−*x*_La*_x_*Fe_12−*x*_Zn*_x_*O_19_ (*x* = 0–0.5, PZTNB-1 is an industrial piezoelectric material based on lead titanate zirconate) did not contain the impurity phases. Due to interfacial doping processes with *x* increasing within the range of 0 to 0.5, a significant improvement in the piezoelectric properties of the composites, as well as a change in their magnetic parameters towards the increase of the magnetic softness could be observed: Piezoelectric coefficients d_33_ = 10–60 and −d_31_ = 2–30 pC/N, piezoelectric voltage coefficients g_33_ = 2–13 and −g_31_ = 1–5 mV m/N, an electromechanical coupling coefficient K_p_ = 0.03–0.13, coercivity of H_c_ = 3–1 kOe, saturation magnetization M_s_ = 50–30, and remanent magnetization M_r_ = 25–12 emu/g. Pre-magnetized and electrically polarized ME ceramics 50 wt.% PZTNB-1 + 50 wt.% Pb_1−*x*_La*_x_*Fe_12−*x*_Zn*_x_*O_19_ (*x* = 0–0.5) had a linear ME effect, and the maximum ME conversion coefficient ΔE/ΔH ~ 1.75 mV/(cm Oe) was achieved at *x* = 0.5.

## Figures and Tables

**Figure 1 nanomaterials-10-01630-f001:**
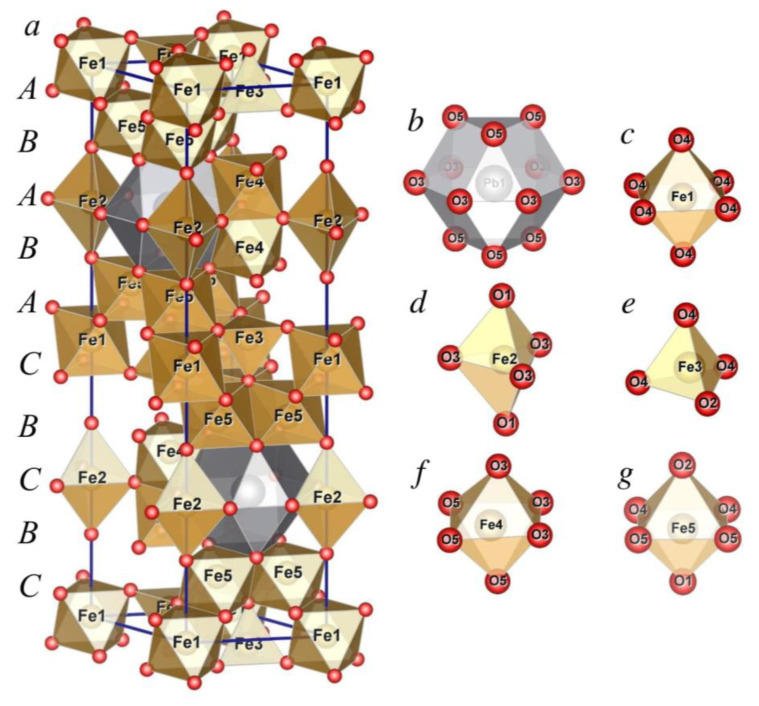
The unit cell of PbFe_12_O_19_ (**a**) and the coordination polyhedra of the cations Pb^2+^ (**b**) and Fe^3+^ (**c**–**g**).

**Figure 2 nanomaterials-10-01630-f002:**
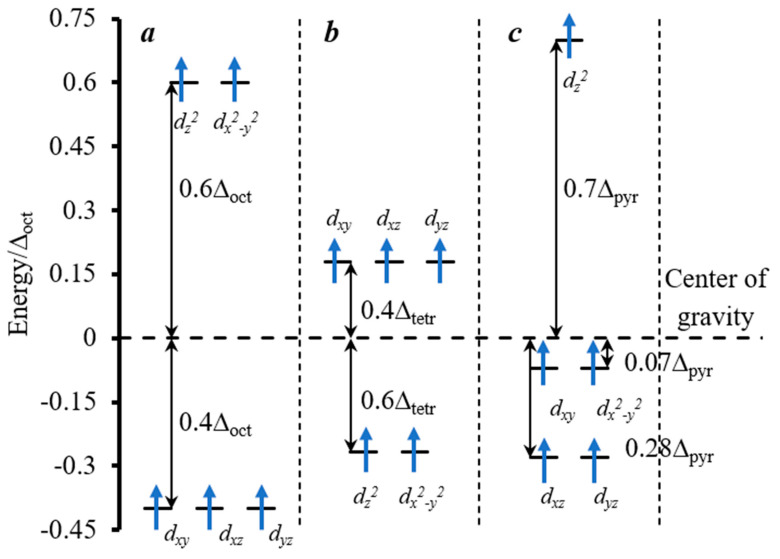
The splitting of the d-sublevel in (**a**) the octahedral, (**b**) tetrahedral, and (**c**) the trigonal-bipyramidal field of the ligands and its population by the electrons for Fe^3+^ (d^5^) in the weak ligand field.

**Figure 3 nanomaterials-10-01630-f003:**
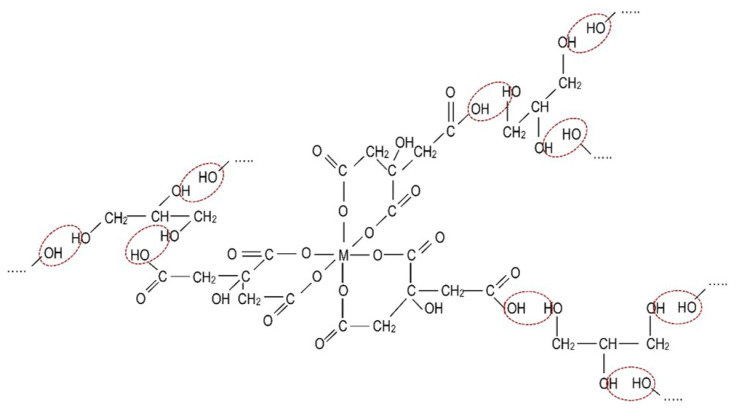
Scheme of the formation of the citrate–glycerin gel.

**Figure 4 nanomaterials-10-01630-f004:**
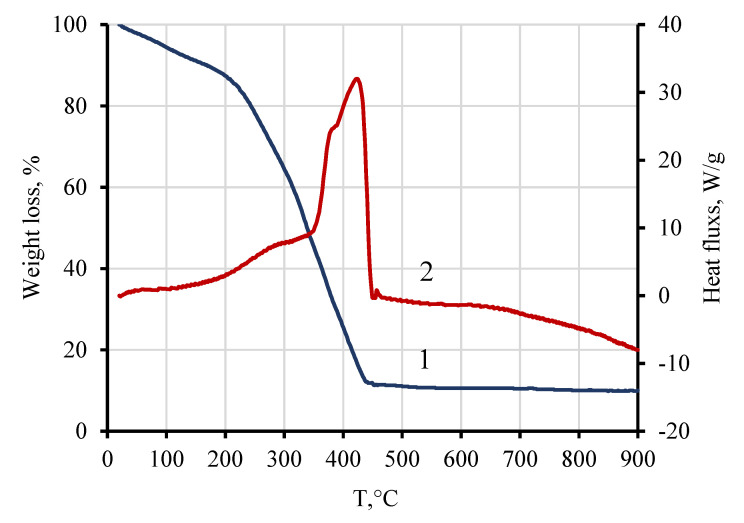
TGA (1) and DSC (2) curves of the xerogel obtained during the synthesis of PbFe_12_O_19_.

**Figure 5 nanomaterials-10-01630-f005:**
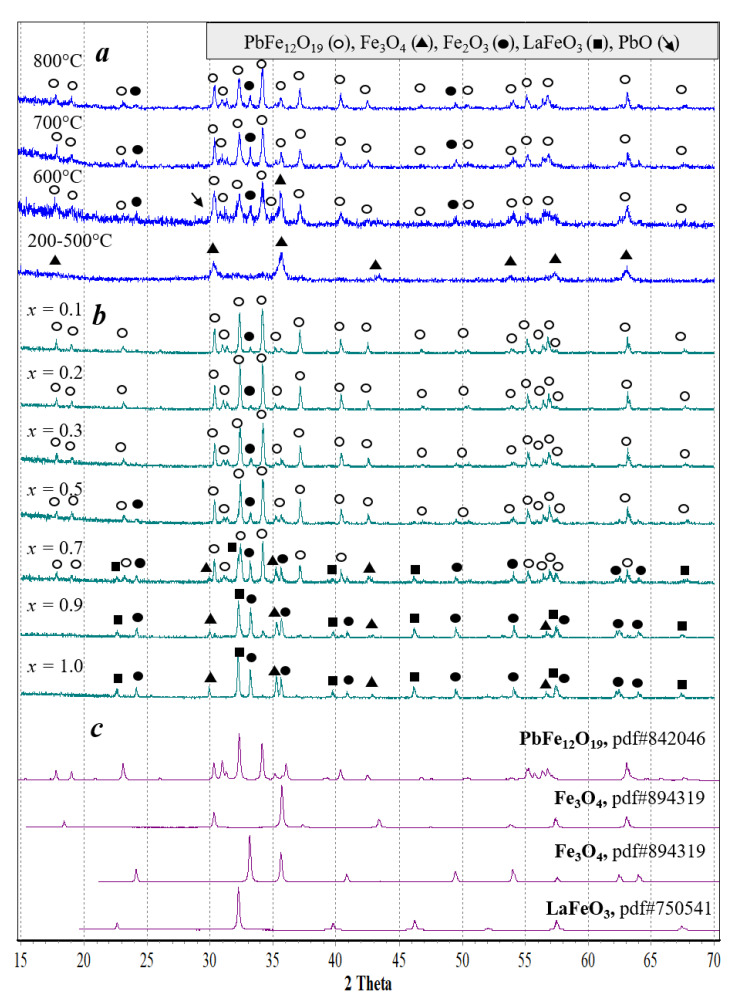
Diffraction patterns of the products obtained by the gel method: (**a**) PbFe_12_O_19_ after the calcination at various temperatures, (**b**) Pb_1−_*_x_*La*_x_*Fe_12−*x*_Zn*_x_*O_19_ after the calcination at 800 °C, and (**c**) diffraction profiles of some oxide phases from the ICDD base (for comparison).

**Figure 6 nanomaterials-10-01630-f006:**
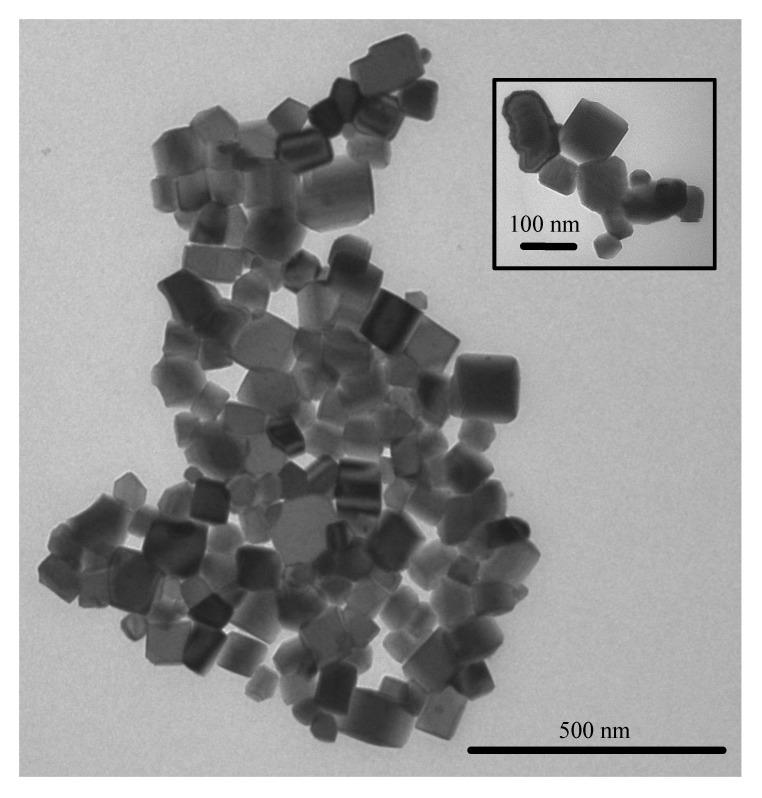
TEM image of Pb_1−*x*_La*_x_*Fe_12−*x*_Zn_x_O_19_ nanoparticles (*x* = 0.1) synthesized by gel processing.

**Figure 7 nanomaterials-10-01630-f007:**
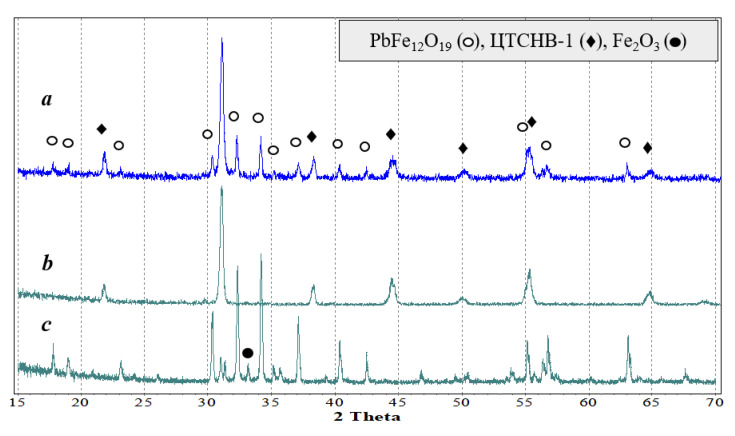
Powder X-ray diffraction patterns (**a**) of composite ceramics 50 wt.% PZTNB-1 + 50 wt.% Pb_1−*x*_La*_x_*Fe_12−*x*_Zn*_x_*O_19_ (*x* = 0.1) and the pure phases: (**b**) PZTNB-1 and (**c**) Pb_0.9_La_0.1_Fe_11.9_Zn_0.1_O_19_.

**Figure 8 nanomaterials-10-01630-f008:**
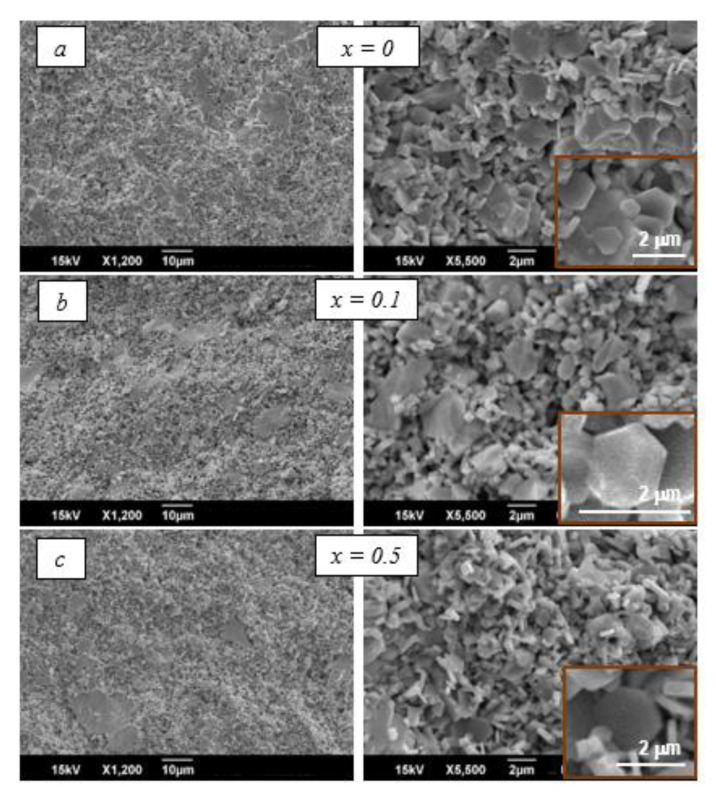
SEM images of ME ceramics 50 wt.% PZTNB-1 + 50 wt.% Pb1−xLaxFe12−xZnxO19 with (**a**) x = 0, (**b**) x = 0.1, (**c**) x = 0.5.

**Figure 9 nanomaterials-10-01630-f009:**
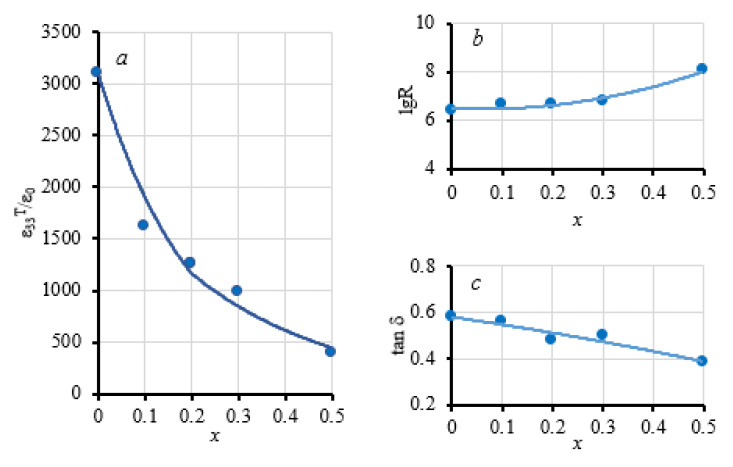
The concentration dependence of the dielectric constant (**a**), the logarithm of the electrical resistivity (**b**), and the dielectric loss tangent (**c**) of composite ceramics 50 wt.% PZTNB-1 + 50 wt.% Pb_1−*x*_La*_x_*Fe_12−*x*_Zn*_x_*O_19_.

**Figure 10 nanomaterials-10-01630-f010:**
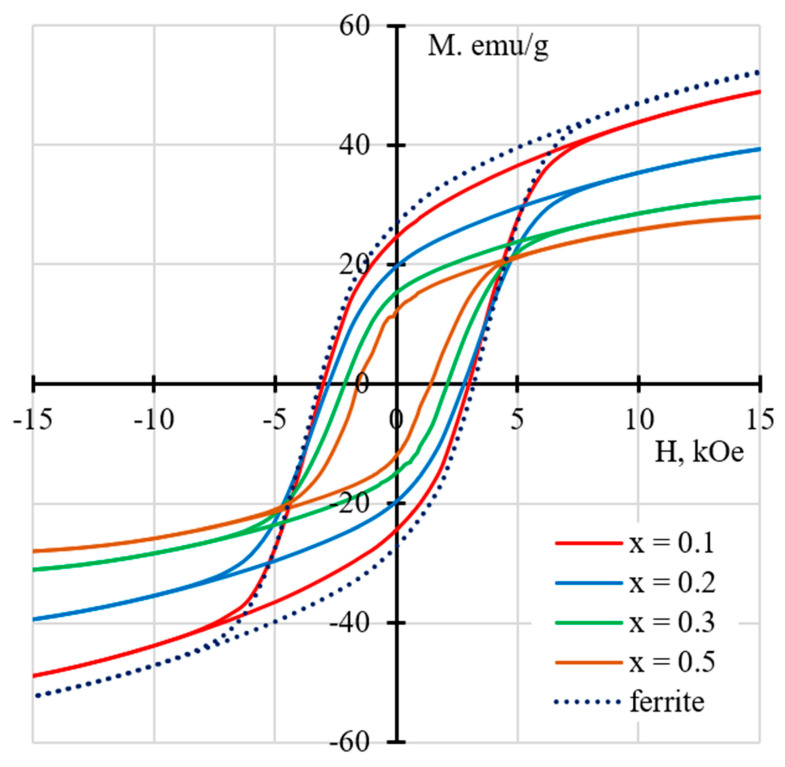
Magnetic hysteresis loops for ME ceramics 50 wt.% PZTNB-1 + 50 wt.% Pb_1−*x*_La*_x_*Fe_12−*x*_Zn*_x_*O_19_ and pure ferrite with *x* = 0.2 (dashed curve).

**Figure 11 nanomaterials-10-01630-f011:**
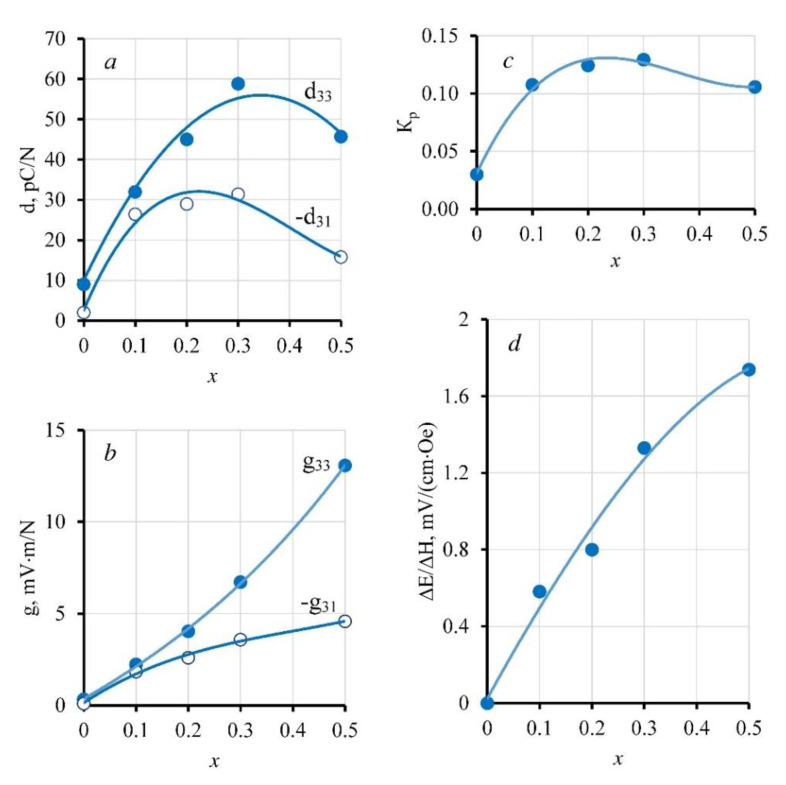
The concentration dependence of the piezoelectric coefficients d_33_ and −d_31_ (**a**), the piezoelectric voltage coefficients g_33_ and −g_31_ (**b**), the electromechanical coupling coefficient K_p_ (**c**), and the ME coefficient ΔE/ΔH (**d**) of composite ceramics 50 wt.% PZTNB-1 + 50 wt.% Pb_1−*x*_La*_x_*Fe_12−*x*_Zn*_x_*O_19_.

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
