# Peer review of "Gel Synthesis of Hexaferrites Pb1−xLaxFe12−xZnxO19 and Properties of Multiferroic Composite Ceramics PZT–Pb1−xLaxFe12−xZnxO19"

_nanomaterials, 2020, doi:10.3390/nano10091630_

Round 1

Reviewer 1 Report

Comments in attached file.

Reviewer 2 Report

In this manuscript, the authors presented the investigation of the synthesis of Pb1-хLaхFe12-xZnxO19 and the different properties of this material. Many characterizations are done on this material. However, the organization of this manuscript is not good and makes it difficult to read and get the points. The following comments need to be considered before the publication.

  1. The introduction part is not well organized. Many literatures is listed, but it is hard to get the key points. I suggested the introduction part should be rewritten and make it more concise.
  2. In the introduction, the authors did not mention clearly what they have done and the importance of their work, which is difficult for a reader to understand the manuscript.
  3. The coercivity decreases in the ferrite within the range of x = 0-0.5 as shown in figure 9. However, the coercivity for the sample with x = 0.5 is still in the range of 1.5-2.0 kOe. It is not magnetically soft.
  4. The conclusion part should be more concise and well organized. It is hard to get the key points of this work by just listing experimental results with long sentences.

Reviewer 3 Report

It is a good paper. Please do some English spell check.

Round 2

Reviewer 1 Report

The authors have improved their paper substantially. I am happy that they have responded to each point that I raised in my earlier report and have included additional data that has definitely improved the quality of the paper. In my opinion, this article can be published in Nanomaterials after the following minor corrections.

  1. I suggest that the authors proofread their manuscript thoroughly. Minor typographical errors can leave a very bad impression on readers. For example, there are mistakes in the author names in reference 34. “Joshia” should actually be “Joshi”, and “Sarkara” should actually be “Sarkar”. Similarly, there are typographical errors in other parts of the manuscript too. In fact, reviewer 3 had already raised this point in his/her earlier report, but I suspect that the authors have not taken this seriously.
  2. At the top of page 6, the authors write that the morphological characteristics were determined using “transparent electron microscopy (TEM)”. The full form of TEM is not “transparent electron microscopy”, it is “transmission electron microscopy”.

Reviewer 2 Report

The authors have answered my questions. I think it is ok for publication.
